# mTOR Signaling Pathway in Cancer Targets Photodynamic Therapy In Vitro

**DOI:** 10.3390/cells8050431

**Published:** 2019-05-09

**Authors:** Sandra M. Ayuk, Heidi Abrahamse

**Affiliations:** Laser Research Centre, Faculty of Health Sciences, University of Johannesburg, P.O. Box 17011 Doornfontein, South Africa; habrahamse@uj.ac.za

**Keywords:** cancers, mTOR, inhibitors, photodynamic therapy

## Abstract

The Mechanistic or Mammalian Target of Rapamycin (mTOR) is a major signaling pathway in eukaryotic cells belonging to the P13K-related kinase family of the serine/threonine protein kinase. It has been established that mTOR plays a central role in cellular processes and implicated in various cancers, diabetes, and in the aging process with very poor prognosis. Inhibition of the mTOR pathway in the cells may improve the therapeutic index in cancer treatment. Photodynamic therapy (PDT) has been established to selectively eradicate neoplasia at clearly delineated malignant lesions. This review highlights recent advances in understanding the role or regulation of mTOR in cancer therapy. It also discusses how mTOR currently contributes to cancer as well as future perspectives on targeting mTOR therapeutically in cancer in vitro.

## 1. Introduction

The Mechanistic or Mammalian Target of Rapamycin (mTOR) pathway incorporates both intra and extracellular signals, and functions as a key regulator of physiological processes including in the growth, metabolism, proliferation, metastasis and malignant transformation of various human tumors [1]. Based on statistics from the Cancer Genome Atlas Pan-Cancer effort, the mTOR signaling pathway was found to be one of the highest mutated genes in 12 cancers analyzed from 3281 tumors. Examples of these cancers include breast, colon, lung, uterine corpus endometrioid, head and neck as well as ovarian [2,3]. mTOR receives signals from its effectors to control the cell function and homeostasis in normal cells. However, in cancer cells, this function is lost. Somatic mutation and gene amplification encode key components leading to the activation of the pathway that enhances cell proliferation and tumor growth [4,5,6,7,8]. mTOR serves as the major growth and survival pathway for cancer pathogenesis and has been an attractive target development of anticancer therapies. mTOR functions in controlling the downstream processes of ribosomes, mRNA, protein synthesis as well as translation. To achieve these functions, they interfere with various signaling pathways including nuclear factor kappa-light-chain-enhancer of activated B cells (NF-kB), phosphatidylinositol-3-kinase (PI3K)/AKT, reticular activating system (RAS), and tuberous sclerosis complex (TSC). When deregulated, they may induce uncontrolled cell growth and proliferation [9]. Furthermore, growth factors such as tyrosine kinase receptors play an important role in the downstream processes within the pathway to enhance biological processes such as angiogenesis, proliferation, metabolism, survival and differentiation [4]. The pathway may therefore be very useful in cancer pathogenesis and disease progression if it is altered and further lead to the development of molecularly targeted treatments that could advance into successful clinical trials [10].

Various inhibitors and signaling components for downstream processes have shown promising results in clinical trials. Clinically, relevant inhibitors target different pathways that present high sensitivity and needs to be studied [11,12,13]. Second-generation mTOR inhibitors have shown improved antitumor activity both in animal models and in vitro. Some previously studied 1^st^ generation inhibitors have shown very little sensitivity including 1st generation rapamycin derivatives (Rapalogs) which have not proven to be very efficient due to their pharmacodynamics. There is still ongoing preclinical and clinical trials to evaluate various targets [14]. Several cancers become resistant to conventional therapies leading to poor prognostics [2,3] and in the effort to enhance therapy and curb resistance, several combination therapies are been investigated [6,15,16].

Photodynamic therapy (PDT) was originally developed about a hundred years ago for the treatment of various tumors and other non-malignant diseases [17]. The treatment mechanism involves the injection of a non-toxic photosensitizer (PS) locally, systemically or topically to a specific lesion accompanied by the absorption of visible light of a particular wavelength in the presence of oxygen from the singlet state to the triplet state as a means of generating cytotoxic reactions [18]. These reactions form reactive oxygen species (ROS) which result in tissue destruction, pathogenic microbes and cell death [19,20] (Figure 1). Photo activation may destroy cancer cells through apoptosis, necrosis or autophagy based on the organelle which the PS has accumulated [21]. PDT specifically targets malignant tumors and destroys the cell with minimal side effects [7]. Photoreactions release oxidant species which may alter the cell, its microenvironment, or even the whole organism. The process involves two types of reaction pathways namely type I (radicals and ROS) and type II (Singlet oxygen) [18] (Figure 1). More oxygen molecules are produced in the singlet state which makes type II more predominant [18]. The action of an ideal PS is based on various factors including PS concentration and localization, amount of energy released, the genetic profile, the dosage administered and wavelength [20]. mTOR has also been demonstrated as a target for PDT in vivo using the lysosomal-based phthalocyanine derivative. This was proven effective in treating 4-Nitroquinoline-1-Oxide (4-NQO) induced murine oral cancer. Velloso, et al. [22] found that the PI3K/Akt/mTOR pathway was inhibited in Human Oral Squamous Cell Carcinoma (OSCC) cells using Aluminum Phthalocyanine (AlPc)-based PDT. Furthermore, Fateye, et al. [23,24] found PI3K pathway inhibitors to significantly enhance the response of PDT [23,24]. Interactions between the mTOR signaling pathway and PDT is under research. This review focuses on targeting mTOR inhibitors in PDT of cancer cells.

## 2. The mTOR Pathway

The mTOR pathway comprises a 289 kDa serine/threonine kinase situated downstream of the PI3K-AKT signaling pathway [25]. mTOR has been revealed to be a major regulator of cell growth, proliferation, migration, differentiation, and survival [25]. Studies have also shown that mTOR is deregulated in most human cancers both upstream via the PI3K-AKT pathway and downstream via the 4E-binding protein 1 (4E-BP1) and Ribosomal protein S6 kinase beta-1 (S6 kinase) pathway, all of which make it a target for tumor suppression [26]. Being the most distorted pathway in human cancers, thePI3K signaling pathway plays a very important role in tumor cell survival and progression. AKT and mTOR are further activated downstream mechanism through the conversion of phosphatidylinositol-4, 5-biphosphate (PIP2) to phosphatidylinositol-3, 4, 5-triphosphate (PIP3) in the cell membrane to induce a cascade of protein phosphorylation (Figure 2). Abnormal activation can enhance tumorigenesis making the pathway a highly attractive target for cancer therapy [27]. mTOR consists of various domains involved in the physiological process, namely the binding or HEAT domain composed of two N-terminals and involved in protein-protein interactions, the FRB (FKPB12-rapamycin binding domain) domain of mTOR which is the binding domain for rapamycin, the FAT and c-terminal FATC (FAT Carboxyterminal) domain present in P13K-related kinases as well as the catalytic kinase domain [28].

Through interactions with nutrients, growth factors and energy stores, mTOR can directly affect cell proliferation and differentiation [29]. Furthermore, mTOR comprises a catalytic subunit of two unique protein complexes, namely mTOR complex 1 (mTORC1) and 2 (mTORC2) [30]. These complexes are unique in their function. mTORC1 is stimulated during cell activation whereby T-cell receptor (TCR) stimulates activation of P13K [31]. This activation is catalyzed by the pyruvate dehydrogenase kinase 1 enzyme (PDK1) [32]. mTORC1 comprises of three mTOR catalytic subunits, namely the regulatory associated protein of mTOR (RAPTOR), mammalian lethal with SEC13 protein 8 (MLST8), and the noncore components PRAS40 and DEP domain-containing mTOR-interacting protein (DEPTOR). When mTORC1 is activated, it phosphorylates the effectors which are major regulators of protein translation including translation-regulating factors ribosomal S6 kinase-1 (S6K-1) and eukaryote translation initiation factor 4E binding protein-1 (4EBP-1) to enhance protein synthesis [31,33,34] (Figure 2).

mTORC2, on the other hand, can be directly activated by P13K [35]. It phosphorylates and activates AKT and other related kinases [36]. Furthermore, through the PI3K-AKT signaling, co-stimulatory signals from cytokines and TCR can also activate the mTOR signaling pathway to further activate the T cells and attain energy supplies [37]. It comprises three proteins, namely the rapamycin-insensitive companion of mTOR (RICTOR), MLST8 and the mammalian stress-activated protein kinase interacting protein 1 (SIN1). Activation occurs through the phosphorylation of AKT at serine-473 [36,38]. Some cells have the same sensitivity to rapamycin [39] but rapamycin selectively inhibits mTOR with more sensitivity to mTORC1 compared to mTORC2 [40]. Studies have shown that mTORC2, as opposed to mTORC1, lacks sensitivity to rapamycin inhibition. Most cancer cells are resistant to the 1^st^ generation mTOR inhibitors (Rapalogs) which particularly target mTORC1 which makes the insensitivity of mTORC2 a possible opening for drug discovery [41].

## 3. The Role of mTOR Inhibitors in Cancer

mTOR inhibitors can be classified into first and second generations depending on their mechanisms and targets. The first generation uses allosteric mechanisms to block the mTOR pathway while the second generation prevents kinase activity in both mTORC1 and 2 using their target ATP binding site. Examples of the 1st generation include the rapamycin and its analogs while the second generation includes AZD8055, Torin1, PP242 and PP30 [42]. Based on some clinical trials mTOR inhibitors are implicated in tumor cells with p53 and PTEN mutations [43]. Three generations of inhibitors has been developed namely Rapalogs (Rapamycin and derivatives), ATP-competitive inhibitors and the Rapalink [44].

Rapamycin also referred to as sirolimus was discovered as an antifungal, immunosuppressive and antitumor compound isolated from Streptomyces hygroscopicus a soil bacterium [45,46]. This drug was initially approved as an anti-host rejecter in 1997 by the food and drug administration (FDA) for kidney transplants [47]. It also functions in many human cancers mainly for the inhibition of signal transduction pathways by forming complexes with peptidyl-prolyl-isomerase FKBP12. These pathways are necessary for cell growth and proliferation [9]. According to Shafer, et al. [48], its anti-angiogenic and proliferative property can be seen in phase II preclinical studies on endometrial cancer cell lines whereby it has a synergistic effect on the paclitaxel. mTOR has been revealed to be the homolog of yeast TOR/DRR genes previously identified in genetic screens their resistance to rapamycin [49]. It has also been identified as a direct target of the complex of FKBP12-rapamycin (FRB domain) [50]. The mechanism of action for rapamycin is based on the binding of mTOR and rapamycin complex FKBP-12 with phosphatidic acid to block the function of mTOR kinase. It attaches to the FRB domain of the mTOR and finally destabilizes the mTOR–raptor–4EBP1/S6K-1 scaffold complex through the binding of mTOR and the complex FKBP-12–rapamycin. These result in dephosphorylation of 4EBP1 and S6K-1 [51,52]. The FRB domain is adjacent to the kinase domain and limits access to substrates to the kinase site [53,54]. However, rapamycin lacks sensitivity in some binding sites making them less sensitive [55].

The therapeutic development of mTOR inhibitors has improved due to their importance in cancer progression and development [56]. Several inhibitors have been approved by the FDA and are already being implemented in the treatment of various human cancers such as breast cancer (everolimus), metastatic renal cell cancer carcinoma (everolimus and temsirolimus), pancreatic neuroendocrine tumors (everolimus) and mantle cell lymphoma (temsirolimus) [57]. Temsirolimus (CCI-779), everolimus (RAD001), and ridaforolimus (MK-8669/AP23573) [6,58,59] have been improved due to their poor aqueous solubility and bioavailability. Studies have shown that rapamycin and its Rapalogs inhibit mTORC2 complex in a way that is independent on time, cell type and dose and based on interaction with newly synthesized molecules of complexes of rapamycin/Rapalogs-FKBP12 and mTOR molecules. This results in further interaction with RICTOR. Studies have shown that the inhibition of components such as RICTOR, RAPTOR, or mTOR significantly reduces the proliferation of cancer cells and offsets progression in the cell cycle [60,61,62]. Overexposure of cancer cells to rapamycin may encourage mTOR binding and inhibit AKT mediated signaling even before the mTORC2 complex is formed [63].

Rapalogs present antiproliferative characteristics in cells that have not been transformed and can efficiently inhibit T-cell proliferation in patients who have undergone transplants [64,65]. They have also shown antitumor responses in benign tumors of TSC [66,67] including lymphangiomyomatosis, renal angiomyolipoma, cardiac rhabdomyoma, facial angiofibroma and retinal astrocytic hamartoma [66]. Reduced efficacy was seen in sporadic cancers and when treatment was stopped [68,69]. Recently they have been approved for the treatment of various tumors including renal cell carcinoma [70,71], postmenopausal hormone receptor-positive advanced breast cancer in combination with exemestane [72], advanced pancreatic neuroendocrine tumors [73], advanced non-functional neuroendocrine tumors of the gastrointestinal tract or lung [74] and relapsed or refractory mantle cell lymphoma [75].

Gulhati, et al. [60] found that the knockdown of mTORC1and 2 mediated in colorectal cancer xenografts in vivo slows down the development of rapamycin sensitive and insensitive cell lines. In addition, the knockdown of mTORC2 increased apoptosis in colorectal cancer cells resistant to rapamycin. Guertin, et al. [76] also found that prostate cancer was mTORC2 dependent when induced in the prostate epithelium by phosphatase and tensin homolog deletion. mTOR is also vital in advanced cancer development and metastatic cancers. It alters the tumor environment to promote metastasis. The hyperactivation of mTOR by RICTOR enhanced cell proliferation in gliomas [77]. Inhibition of mTOR may also improve the way chemotherapeutic agents respond in advanced diseases. Patel, et al. [78] found that the inhibition of mTOR prevented the distribution of cancer cells to lymph nodes slowing down angiogenesis in head and neck cancer.

Everolimus was approved as an oral mTOR inhibitor for advanced renal cell cancer. It is also known for its anti-proliferative and angiogenic activity in human cancers [79,80]. This includes metastatic pancreatic neuroendocrine tumors, metastatic renal cell carcinoma, advanced estrogen receptor (ER)-positive [79] and human epidermal growth factor receptor-2 (HER2)-negative breast cancer [80]. Studies have recorded an improvement in cancer when rapamycin and its Rapalogs were used in combination with either standard chemotherapy, hormonal therapy, or alone. A current study found significant progression-free survival (PFS) when patients with HER-2 advanced stage of breast cancer, pre-treated with taxane and trastuzumab were administered both everolimus together with trastuzumab and vinorelbine [81]. Another breast cancer study by Hurvitz, et al. [82] found the combination of everolimus and paclitaxel and trastuzumab promising. Temsirolimus (Torisel^®^), was approved by the FDA as the 1^st^ rapamycin analog to be used for the treatment of cancer cells. It is an intravenous injection which when injected in vivo becomes converted into rapamycin. Studies have shown an increase in progesterone mRNA and the inhibition of endoplasmic reticulum mRNA expressions when administered with bevacizumab or in combination with other chemotherapeutic agents for treating endometrial cancer cell lines [48,83]. In addition, Tinker, et al. [84] found positive results after using temsirolius for a preliminary phase II study in patients with metastatic cervical cancer. The drugs were effective when administered together with paclitaxel/carboplatin for treating stage II/IV patients with clear cell adenocarcinoma on a clinical phase II trial [85].

Not all studies have however proven positive outcomes with the drug. Behbakht, et al. [86] found decreased activity of temsirolimus with the drug efficacy failing in patients with primary peritoneal cancer or persistent/recurrent epithelial ovarian cancer. These results still need to be investigated by a phase III trial. Another inhibitor, Ridaforolimus (MK-8669/AP23573), a non-rapamycin pro-drug available in both intravenous and oral formulations has been evaluated in combination or as monotherapy on various cancers including breast, prostate, endometrial, sarcomas and non-small cell lung cancer [16]. It has had a 33% response rate when administered to patients with advanced endometrial cancer [87]. A phase two II study showed a partial response rate of 7.7% in advanced or recurrent [88]. Side effects of this drug include low toxicity with dose dependent skin rashes and mucositis [89] as well as hypertriglyceridemia, hypercholesterolemia, nausea, fatigue, anemia, and neutropenia [90]. In addition, sirolimus and temsirolimus present intense pulmonary toxicity. Other side effects include the risk of secondary lymphoma, interstitial lung disease, and the reactivation of latent infections. However, these are rare [91].

Even though Rapalogs are still been used in clinics as opposed to ATP-competitive inhibitors which have not yet been approved, and Rapalink still being developed and subject to experimentation. Several shortcomings of Rapalogs [55] have made the 2nd generation inhibitors better [92,93]. ATP-competitive inhibitors are the second-generation inhibitors. They inhibit both mTORC1 and 2 by blocking the kinase domain [94,95]. As opposed to the Rapalogs, inhibition is intense with blocking of the P13K from their kinase similarity [94]. Rapalink is the third-generation inhibitors designed to curb resistance mutations in both the rapalog and ATP-competitive inhibitors. These inhibitor crosslinks with kinase in the same molecule [96].

## 4. The Role of mTOR Pathway in Cancer Therapy

A major development has taken place in the last few years to understand the role of mTOR in cancer development and progression. mTOR and/or its components have been implicated in various genetic mutations of human malignant diseases [97,98,99]. Mutations of closely related pathways have enhanced mTOR signaling in cancers [59,95,100]. Currently, human cancer genome databases are being mined to aid identification of activated mTOR mutations [101]. Transmitted extracellular signals go through various pathways but P13K/AKT/RAS/RAF/MEK/MAPK are the most common and highly characterized. Using the same mechanism to activate PI3K/AKT/mTOR pathway has presented enhanced tumor progression and poor survival response to patients with different types of tumors [60,102]. Due to its vital function in cell growth and proliferation, its components have been increasingly used as potentials for therapeutic targets. Molecular approaches have been used to establish the role of the components of the mTOR pathway in cancer development. Components of the mTOR pathway have also been activated in various neuroendocrine tumors with a tendency of releasing bioactive products [103,104].

mTORC1 induces nucleotide and protein synthesis to regulate cellular growth via ribosome biogenesis, inhibits autophagy, protein, and nucleotide synthesis. When conditions are favorable, they sense environmental signals such as nutrients and growth factors to initiate cell growth but if conditions become unfavorable in cases of acidity and hypoxia, mTOR activity is inhibited [105,106]. When these pathways are activated, they inhibit mTORC1 through phosphorylation and inhibition of the protein complex, TSC 1 and 2. Mutation of the TSC genes causes TSC disease with benign tumors found in the brain, kidneys, heart, lungs and liver [107]. Activation could lead to the loss of phosphatase and tensin homolog (PTEN). This uncouples mTORC1 activation from growth factor signaling such as mutations of liver kinase B1/serine/threonine kinase 11 (LKB1/STK11) in nutrient-deprived vascular tumors but allows activation of mTORC1. The mutation of P53 inhibits bioenergetics processes and cell cycle arrest uncoupling DNA damage [95]. In addition, it could lead to hyperactivation of S6K-1, 4EBP1 and eukaryotic translation initiation factor 4E (eIF4E), as well as cancer growth through the activation of lipid and protein biosynthesis. Upon activation of S6K-1, 4EBP1 and other substrates are phosphorylated enhancing cell proliferation and growth from an anabolic cellular response [31,35,108]. Through the stimulation of the activity and expressions in small GTPases such as Rac1, cdc42 and Rho to control the activities of the actin cytoskeleton and motility [58,109]. Furthermore, S6K-1 and 4EBP1 mediated by mTORC1 extend vital roles in focal adhesion proteins phosphorylation including paxillion, p130 Cas and focal adhesion kinase as well as reorganization of F-actin [110].

Hyperactivation of mTORC1 results from mutations of mTOR or upstream effectors. This occurs in sporadic cancers [111,112,113]. Furthermore, in hamartoma syndromes, they are characterized by the growth of benign tumors and mutations in tumor suppressor genes [114]. The association of phosphorylated mTOR with AKT signaling and acquired cisplatin resistance affects primary platinum resistance and sensitivity to ovarian cancer cells [115]. Furthermore, these inhibitors restore chemosensitivity to platinum derivate both in vitro and in xenograft models [16,116]. Gulhati, Bowen, Liu, Stevens, Rychahou, Chen, Lee, Weiss, O’Connor and Gao [109] found that mTORC1 was associated with motility, metastasis, and epithelial-mesenchymal transition in colorectal cancer. mTORC1 activity has also been studied in breast cancer and gliomas [77,117]. Despite all these discoveries, more research needs to be conducted to understand how these components are regulated. Gulhati, Bowen, Liu, Stevens, Rychahou, Chen, Lee, Weiss, O’Connor and Gao [109] found that using oxaliplatin in colorectal cancer cells induced apoptosis as a result of the knockdown of mTORC1 and 2. mTORC1 is found to be associated with the transport hormone and, peptide-containing vesicles. They also regulate intestinal hormones which play a vital role in the gastrointestinal tract as well as other secreted neuroendocrine tumors to regulate neurotensin [104].

mTORC2, on the other hand, is activated via growth factors [118]. It phosphorylates and activates the AGC protein kinases including SGK1 (Ser422) and AKT (Ser473). Inhibiting mTORC2 activities will enhance the antitumor effect in several preclinical trials [61,69,76,119,120]. Varied molecular adjustments occur with this pathway which may suggest strategic therapy against cancer cells if targeted. The onset of cancer is provoked by enhanced cell growth and immune escape due to a build-up of genetic and epigenetic changes. Therefore an approach to cancer therapy would be to prevent these changes [121]. Tumor heterogeneity, as well as cellular resistance, are some of the hindrances to targeted cancer therapies. Activation of bypass mechanisms as well as making secondary reforms in the target are resistance mechanisms which have been identified [122]. Nonetheless, most of the targeted treatments have not been beneficial in the long run despite all the preclinical trials.

### mTOR Signaling Pathway and PDT

The PS are composed of natural occurring macrocycles including hemoglobin, vitamin B12 and chlorophyll. These compounds consist of nitrogen, oxygen, or sulfur atoms locked in a hollow ring containing metals such as iron or magnesium. Currently, PDT makes use of plant extracts to complex synthetic macrocycles. These different agents can selectively target and accumulate in the tumor. The widely investigated PS includes tetrapyrroles such as bacteriochlorins, chlorins, porphyrins, and phthalocyanines [123]. To improve efficacy, clinical considerations have been given to other compounds such as synthetic dyes and targeted therapies which use various drug delivery systems to improve the penetration of light. The fate and effectiveness of PDT on tumors are based on the oxygen concentration, wavelength, types of photosensitizer and the genotype of the cell. This can affect certain organelles and specific target tissue [20]. Dual-specificity of the PS would depend on accumulation and localization of the PS in diseased tissue. The PS if hydrophobic accumulates in the mitochondria and endoplasmic reticulum, other polar compounds may Golgi apparatus, lysosomes and plasma membrane [124].

PDT down-regulates AKT-mTOR pathway because of ROS production (Figure 3). In modern oncology, a combination of different therapeutic modalities with non-overlapping toxic effects are strategies used to improve the therapeutic index of treatment. Combination therapies target different disease pathways, which represents an alternative approach that might offer potential advantages over a single therapy.

Few studies have shown these combining effects on PDT. Kraus, et al. [125] found that combining P13K/mTOR inhibitors (BYL719, BKM120, and BEZ235) with verteporfin-PDT to synergistically enhance PDT response with BEZ235 presenting the strongest. Antiapoptotic inhibition of the Bcl-2 family protein Mcl-1 and P13K pathway was critical. Fateye, et al. [24] assessed the effect of combination of P13K/mTOR inhibitor (BEZ 235 (BEZ)) on PDT efficacy using prostate tumor (PC3) and SV40-transformed mouse endothelial cell lines (SVEC-40) and found that the sub-lethal PDT was enhanced in both cell lines. Combination of PDT with pan-PI3/ mTOR kinase inhibitor LY294002 (LY) also enhanced PDT effect with PC3. However, it produced a synergistic effect in SVEC-40. In contrast, Sasore and Kennedy [126] found that there are some combinations of PI3K/AKT/mTOR pathway inhibitors, which actually interrupt developmental angiogenesis due to their additive or synergistic effect. Tuo, et al. [127] used human SZ95 sebocytes to find out the potential pharmaceutical effect of combining ALA-PDT and rapamycin through the mTOR pathway and found that cell growth was suppressed, protein levels of P-mTOR, and P-Raptor were reduced as well as lipogenesis. Their study concluded that rapamycin enhanced aminolevulinic acid hydrochloride (ALA)-PDT in SZ95 cells. mTOR inhibition can induce autophagy in various ways: direct induction, pre-condition cells, or by stressor induction. A study by Weyergang, et al. [26] using colon adenocarcinoma cell line and amphiphilic endolysosome-localizing photosensitizer Al(II) phthalocyanine chloride disulfonic acid (AlPcS(2a)) showed that targeting mTOR signaling pathway in PDT caused partial loss of both total and phosphorylated mTOR in both tumor xenografts and cultured cells in vitro and in vivo. According to Weyergang, et al. [26] combining rapamycin potentiates cytotoxicity in vitro post-PDT. The interest in the combination of PDT and other therapeutic modulates in cancer treatment is to provide a platform for potential treatment options and limited adverse effects of chemotherapy since PDT does not have the inherent dose-limiting toxicity [128]. Combination therapies are aimed at increasing responses, improving patient tolerability, decreasing drug dosages and the emergence of drug resistance [129]. Combined effects of PI3K/AKT/mTOR and PDT as a treatment regimen for cancers still needs further investigation.

## 5. Perspective

Despite its promising minimal and non-toxic side effects, it is still unlikely that administering conventional chemotherapies and/or inhibitors alone will completely cure cancer. There are still challenges in cancer therapy including the activation of other proliferation signaling pathways, treatment-resistant mutations as well as the intramural heterogeneity of mTOR activities. Inhibitors alone have failed to induce tumor regression but are seen as cytostatic causing disease stability rather than death [130]. Another limitation might provide negative feedback loops in the mTOR pathway which have limited the efficiency of these Rapalogs. Taking into high consideration the level of toxicity, combined therapies would be the way forward. New generation inhibitors are being produced which can prevent the catalytic activity of both mTORC1 and mTORC2 complexes and enhance therapeutic indexes.

## Figures and Tables

**Figure 1 cells-08-00431-f001:**
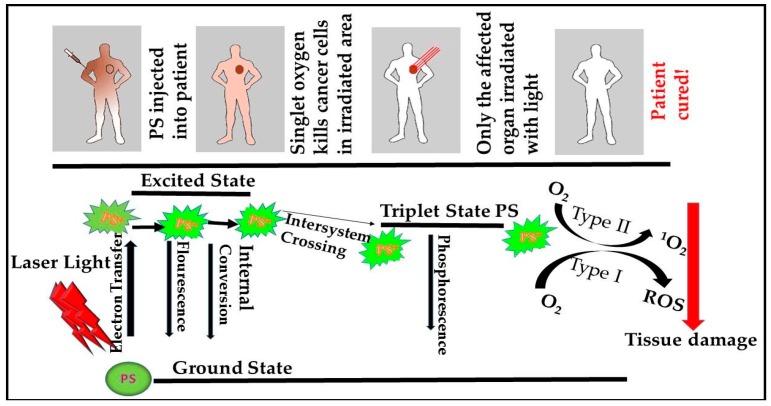
Schematic model of the Mechanism of Photodynamic Therapy (PDT), excitation and relaxation of a photosensitizer, and type I and type II photoreactions. Photosensitizers (PS) after an application as cream or injected become activated by light at specific wavelengths in the presence of oxygen (O_2_). When activated they become excited and move from the singlet state to the triple state generating cytotoxic reactions. Some of the phytophysical reactions include electron transfer, fluorescence, internal conversion, intersystem crossing, and phosphorescence. These reactions directly generate singlet oxygen (^1^O_2_) or indirectly, reactive oxygen species (ROS) resulting in tissue damage and cell death [18].

**Figure 2 cells-08-00431-f002:**
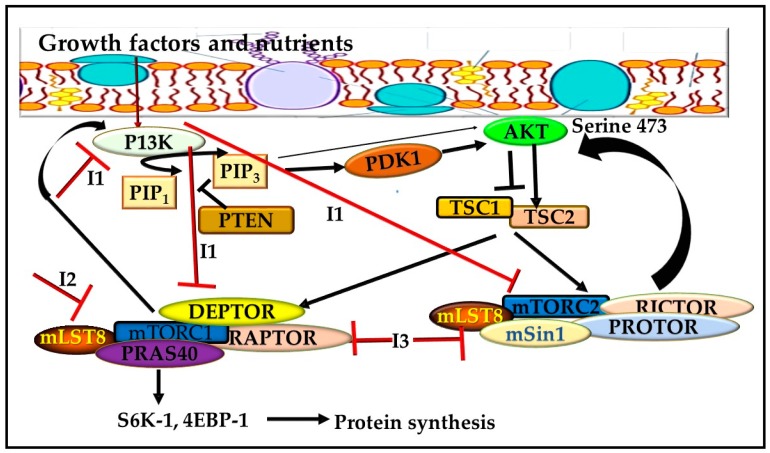
mTOR Signaling pathway. Activation of P13K phosphorylates phosphatidylinositol 4,5-biphosphate (PIP2) to form phosphatidylinositol-3,4,5-triphosphate (PIP3). Phosphatase and tensin homolog deleted on chromosome 10 (PTEN) regulates the function of PIP3. PIP3 prompts the activation of downstream processes such AKT, which transmits signals to effectors including mTOR complexes to enhance cellular processes. The mTORC1 is stimulated during cell activation whereby the T-cell receptor (TCR) stimulates the activation of P13K. mTORC1 comprises of three mTOR catalytic subunits, namely the regulatory associated protein of mTOR (RAPTOR), mammalian lethal with SEC13 protein 8 (MLST8), as well as the noncore components PRAS40 and DEP domain-containing mTOR-interacting protein (DEPTOR). mTORC2 comprises also of three proteins – the namely rapamycin-insensitive companion of mTOR (RICTOR), MLST8, and the mammalian stress-activated protein kinase interacting protein 1 (SIN1). Activation of mTORC2 occurs through the phosphorylation of AKT at serine-473 while that of mTORC1 when activated, phosphorylates the effectors which are major regulators of protein translation including translation-regulating factors ribosomal S6 kinase-1 (S6K-1) and eukaryote translation initiation factor 4E binding protein-1 (4EBP-1) to enhance protein synthesis.

**Figure 3 cells-08-00431-f003:**
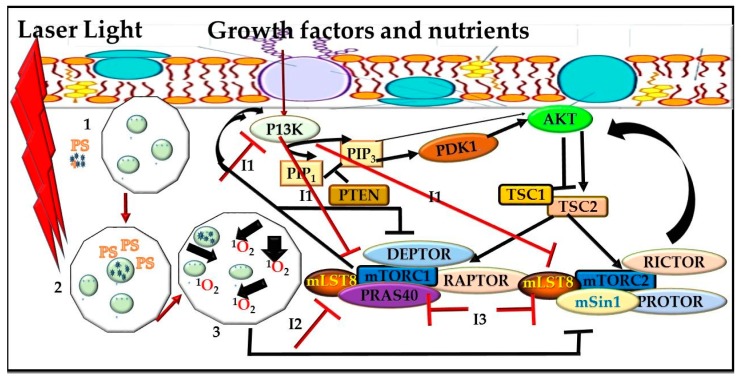
Proposed mechanism between mTOR Signaling Pathway, Inhibitors and Photodynamic Therapy (PDT). PDT down-regulates AKT-mTOR pathway because of reactive oxygen species (ROS) production. 1. Photosensitizer is injected into a targeted tumor. 2. Laser light is emitted at a particular wavelength. 3. Cells become activated and release reactive oxygen species, which results to tissue destruction and cell death. Interaction with inhibitors phosphoinositide 3-kinase (I1), rapamycin (I2) and mTOR kinase (I3) to enhance cell death through the P13K/AKT-mTOR pathway.

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
