# Peer review of "mTOR Signaling Pathway in Cancer Targets Photodynamic Therapy In Vitro"

_cells, 2019, doi:10.3390/cells8050431_

Round 1

Reviewer 1 Report

The review manuscript titled "mTOR Signaling Pathway in Cancer Targets Photodynamic Therapy in vitro" by Ayuk, SM and Abrahamse, H describe various roles of mTOR in cancer biology. The review manuscript covers the mTOR pathway,  role of mTOR inhibitors in cancer, role of mTOR pathway in cancer therapy, mTOR signaling pathway and photodynamic therapy. Following are my comments to improve the manuscript:

1.     A brief historical perspective of mTOR and PDT may be useful.

2.     A cartoon/schematic to depict how PDT works is required for complete understanding of the review.

3.     While the review is useful in the field of mTOR and cancer biology, the only concern I have is with respect to the usage of English and several typos and grammatical errors. These needs to be fixed before accepting the manuscript. I will give couple of examples here:

a)     Line 26, change "3.281" to "3,281"

b)    Line 33. Correct crosstalk

c)     Line 33-37. This is one very long sentence that fails to give a clear message. Please correct and write in simple sentences.

d)    Line 55. Correct repetition.

e)     Line 82. Incomplete sentence.

f)     Line 88. Syntax error.

g)     Line 105.  "mTOR selectively inhibits rapamycin" is completely wrong. Rapamycin inhibits mTOR.

h)    Line 283. Is it type 11 or type 2.

i)      Line 311. It is not clear, why all the author names were written to make a point.

j)      Throughout the manuscript different fonts were used that causes a lot of distraction.

k)    Shown above are not the exhaustive list of examples but only some to point to the authors.

4. Lines 277-279: There is lot of repetition from the earlier sections.

Author Response

REVIEWER 1

Suggestions and comments to the Author

The review manuscript titled "mTOR Signaling Pathway in Cancer Targets Photodynamic Therapy in vitro" by Ayuk, SM and Abrahamse, H describe various roles of mTOR in cancer biology. The review manuscript covers the mTOR pathway, role of mTOR inhibitors in cancer, role of mTOR pathway in cancer therapy, mTOR signaling pathway and photodynamic therapy. 

Following are my comments to improve the manuscript:

Corrections made to manuscript (Cell-486215)

We appreciate your decision to consider our paper as one of the peer reviewed in the journal of CELLS. We welcome the opportunity to reply to your comments. Most of the information requested has been highlighted in red.

Point 1. A brief historical perspective of mTOR and PDT may be useful.

Author’s Response. A brief perspective of mTOR and PDT has been included with figures 1, 2 and 3 .1) The mechanism of PDT 2) The mTOR Pathway and 3) The interaction of PDT and the mTOR pathway.

Point 2. A cartoon/schematic to depict how PDT works is required for complete understanding of the review.

Author’s Response: A schematic model has been included as figure 1.

Point 3. While the review is useful in the field of mTOR and cancer biology, the only concern I have is with respect to the usage of English and several typos and grammatical errors. These needs to be fixed before accepting the manuscript. I will give couple of examples here:

Authors Response: Thank you. The whole manuscript has been thoroughly edited.

a)     Line 26, change "3.281" to "3,281"

Authors Response: Corrected

b)    Line 33. Correct crosstalk

Authors Response: Corrected in point c

c)     Line 33-37. This is one very long sentence that fails to give a clear message. Please correct and write in simple sentences.

Authors Response: Corrected and sentence rephrased to…. “mTOR functions in controlling the downstream processes of ribosomes, mRNA, protein synthesis as well as translation. To achieve these functions, they interfere with various signaling pathways including nuclear factor kappa-light-chain-enhancer of activated B cells (NF-kB), phosphatidylinositol-3-kinase (PI3K)/AKT, reticular activating system (RAS), and tuberous sclerosis complex (TSC). When deregulated they may induce uncontrolled cell growth and proliferation” [9].

d)    Line 55. Correct repetition.

Authors Response: Deleted

e)     Line 82. Incomplete sentence.

Authors Response: Addressed. Rewritten as “ binding or HEAT domain composed of two N-terminals and involved in protein-protein interactions”

f)     Line 88. Syntax error.

Authors Response: Corrected

g)  Line 105.  "mTOR selectively inhibits rapamycin" is completely wrong. Rapamycin inhibits mTOR.

Authors Response: Corrected

h)    Line 283. Is it type 11 or type 2?

Authors Response: Type 2. Rewritten as Type II

i)       Line 311. It is not clear, why all the author names were written to make a point.

Authors Response: This has been corrected.

j) Throughout the manuscript different fonts were used that causes a lot of distraction.

Authors Response: The required font as per the journal template has been used on the highlighted manuscript.

k)    Shown above are not the exhaustive list of examples but only some to point to the authors.

Authors Response: The whole document has been thoroughly edited for errors.

l)        Lines 277-279: There is lot of repetition from the earlier sections.

Authors Response: Thank you. Repetitions have been deleted.

Reviewer 2 Report

The Review require a very profiund English editing. Also organization of the material:less background and more the combination as suggested by the title. 

Author Response

REVIEWER 2

Comments and Suggestions for Authors

Point 1. The Review require a very profiund English editing. Also organization of the material: less background and more the combination as suggested by the title.

Authors Response: Thank you.  The manuscript has been thoroughly edited and additional information included in the combination as suggested.

Reviewer 3 Report

The idea of summarizing mToR as a target for photodynamic therapy is interesting but the text does not support the notion suggested in the title. Extensive elaborations have been made to describe the mToR pathway for which there is a huge body of literature, however, the concept of PDT and its connection to mToR is only brought in in the last paragraph. This is contrary to the expectations of the readers who would expect most information on this aspect according to the title. Additionally, the authors see to draw on only 2 references #124 and 125 to substantiate their analysis on this aspect.

The language is not clear and the text is littered with numerous grammatical mistakes. One representative example exists on page 1 between lines 33 and 37: "It crosstalk various signaling pathways...."

A reorientation of the focus of the article coupled with correction of the grammatical errors may make it publication worthy.  

Author Response

REVIEWER 3

Comments to the Author

Point 1. The idea of summarizing mToR as a target for photodynamic therapy is interesting but the text does not support the notion suggested in the title. Extensive elaborations have been made to describe the mToR pathway for which there is a huge body of literature, however, the concept of PDT and its connection to mToR is only brought in in the last paragraph. This is contrary to the expectations of the readers who would expect most information on this aspect according to the title.

Authors Response: More literature has been included on PDT and a schematic diagram to illustrate the mechanism (Figure 1).

Point 2. The authors see to draw on only 2 references #124 and 125 to substantiate their analysis on this aspect.

Authors Response: Additional references and diagrams illustrating the mTOR pathway (Figure 2) and the interaction between mTOR pathway and PDT (Figure 3) to substantiate this analysis have been included.

Point 3. The language is not clear and the text is littered with numerous grammatical mistakes. One representative example exists on page 1 between lines 33 and 37: "It crosstalk various signaling pathways...."

A reorientation of the focus of the article coupled with correction of the grammatical errors may make it publication worthy. 

Authors Response: Thank you. The manuscript has been edited thoroughly. Additional information on the combination therapy has been included.

Round 2

Reviewer 2 Report

The review is ready to be published

Cells EISSN 2073-4409 Published by MDPI AG, Basel, Switzerland RSS E-Mail Table of Contents Alert
Back to Top